# Leadership Development: Exploring Relational Leadership Implications in Healthcare Organizations

**DOI:** 10.3390/ijerph192315971

**Published:** 2022-11-30

**Authors:** Evangelia Maritsa, Aspasia Goula, Alexandros Psychogios, Georgios Pierrakos

**Affiliations:** 1Department of Business Administration, School of Administrative, Economics and Social Sciences, University of West Attica, 12243 Athens, Greece; 2Birmingham City Business School, Birmingham City University, Birmingham B4 7BD, UK

**Keywords:** leader–member exchange, relational identity, trust, organizational silence, organizational citizenship behavior, healthcare, dynamics, leadership, development

## Abstract

(1) Background: Relational Leadership Theory (RLT) has been gaining rising attention for the past 20 years with studies investigating multiple implications and practices of relationships within organizations. Yet, less attention has been given in healthcare settings. By virtue of the emerging need to move beyond exploring the quality of relationships and to move towards the exploitation of relational dynamics that influence leadership development in healthcare organizations, this study explores both the dyad relationships and the context in which those occur. With recent attention directed to the implementation of human-centered practices and the creation of effective networks to bring desired results, RLT is called on to advance this agenda within healthcare organizations. (2) Material and Methods: Research articles that examined leadership theories over the past thirty years were selected from computerized databases and manual searches. (3) Results: It is argued that the way and context in which relationships are formed between leaders and members is a social process that, in turn, shapes the effectiveness of the management of those organizations. Leadership is not rank—it is the relationship with the relational dynamics that play in the same context, creating evolutionary organizational processes. (4) Conclusions: This paper challenges leadership theory one step further. Exploring an organization through relational leadership theory is much like wearing the lens of ‘cause and effect’ in leadership behavioral studies. Therefore, this study contributes to this direction with a robust co-examination of relational dynamics that take place in the healthcare sector, showcasing a broader framework in which relational leadership is germinated and influences its outcomes.

## 1. Introduction

Although interpersonal relationships have always been an important theme to explore in the organizational literature, the relational perspectives and dynamics along with the context within which those emerge and occur are part of a recaptured knowledge base in leadership development [1,2,3]. Nowadays, leaders face one of the biggest challenges, which is to make people and organizations adaptable and resilient in the face of increasingly dynamic and demanding environments (e.g., bioevents, market unpredictability, abrupt change of work conditions) [4]. For instance, the COVID-19 pandemic has unfolded a disruptive impact on the healthcare sector, and that impact affected businesses and society, respectively [5], underlying the need for leadership development at all levels. In this complex ecosystem, members who constitute the organization’s social capital need to view their everyday practices through the lens of relational leadership theory in order to evolve in the workplace [6,7]. As mentioned in Ulmer’s [8] research, in times of crisis where high levels of stress exist, leadership should focus primarily on people rather than hardware because it is the people that matter most, and leaders’ competencies in the complex market context resemble the pressures and doctrines that are met on the battlefield. Therefore, human relational approaches may provide a subtle understanding of leadership practices that are crucial, especially in healthcare settings where unpredictability coincides with science, and the rational methods of healthcare professionals, and this coincidence affect relationships, their interdependencies, and the effectiveness of management [3,9,10].

There is an emphasis placed on leadership development in healthcare settings due to the ongoing health and economic crisis related to the current health crisis (COVID-19) [2,11,12,13] and the projected rise of bioevents in the years to come when more than two-thirds of the general population may experience exposure to traumatic events (natural disasters, violence, terrorism, etc.) during their life cycle [14,15]. Therefore, a reconfiguration of work socialization, communication, efficiency, and productivity in healthcare settings is needed [16,17] as a reform to the leadership methods that will be adjusted to the new context. Several studies in the past years have mainly focused on the cultural mosaic that constitutes a healthcare setting, for instance, cultural differences, bureaucracy, learning systems, and communication between the clinical directorate, nursing directorate, and administrative directorate, thus viewing leadership through the lens of organizational structures and forms [18,19,20,21]. Other studies focused deeper on human entities, viewing leadership as a leaders’ issue (leader-centered) or mainly as a connection that is based on unidirectional communication (top-down) [21,22,23,24,25]. Even more, since leadership is a dynamic process that exceeds individual capabilities and is viewed from the standpoint of interactions (bonds, tensions, brokerages, dynamics, etc.) [4,26,27], healthcare organizations are considered systems of higher relational dynamism and wider systems of societal interdependencies [9]. Hence, healthcare organizations constitute a rich environment to study human relationships and interactions [28] and exploit their potential to improve organizational effectiveness in the healthcare sector. While there is evidence that leadership is relational [1,3,7,29], very little research focuses on the relational environment of the specific social context from which interactions where healthcare professional relationships are born [24,25,30], as the majority focuses either on this or relationships with patients [16], thus viewing leadership outcomes either from the standpoint of results with patients or straight from outcomes [31].

This research ‘imbues’ current relational-centered studies [30,32,33,34] with a wealth of new insights regarding the system’s dynamics [9,20] by which relational leadership is developed. By putting leader–member exchanges under the microscope, this research contributes to the overall knowledge of how the actors foster interdependences, creating ‘bridges’ with leadership outcomes in the hierarchical healthcare system [31].

## 2. Review of the Literature

### 2.1. Relational Leadership Theory (RLT)

From World War II and thereafter, the concept of relational-oriented behavior has been discussed in traditional management as an important human social aspect [35]. This aspect has been discussed based on hierarchical systems in which leaders raise the salience of organizational values with their behaviors, display high involvement in tasks, link their goals to group perspectives, and support their subordinates [35,36,37,38]. Social exchange theory in the workplace offers an explanatory framework for the interdependencies that occur from the interactions that take place in the local social context of healthcare organizations [39].

Relational leadership is a toolkit that contains the context of interactions by which leadership is developed and enabled, and it also contains the outcomes of the bidirectional relationships (leadership outcomes) created. Therefore, the importance of redirecting emphasis away from the individual leader and towards the system phenomenon that is leadership [4,40] is a key thought behind this research. RLT is a theory that views leadership as a process where both stakeholders (e.g., dyadic leader–member relationships) and the environment around them (social environment, organizational functions, etc.) evolve to either expand the space of their potential influence or shrink the space of their potential influence [7]. Therefore, leadership is inherently relational and will be seen as a process of organizing in a specific context that comprises the web of human interconnections that are valued [7,41].

### 2.2. Defining the Dynamics through Which Leadership Is Developed

#### 2.2.1. Leader–Member Exchange (LMX)

In organizations, human relationships begin with the first major relationship: the leader–member relationship. Therefore, healthcare practitioners and their relationships may be thought of as ‘entities’ (individuals) that interact in dyads (leader–member) in a specific social context, influencing each other, co-producing, and co-constructing on a relationship-based approach [7,29,42], portraying leadership as socially constructed. The level of analysis is interpersonal and placed within the dyadic context (leader–member) [43]. This relationship, given the name Leader–Member Exchange Theory and explained by researchers Fred Dansereau, George Graen, and William Haga [44], who are the fathers of this theory that originated back in 1975, introduces the concept that interpersonal relationships can be increased. Therefore, one can depict LMX exchanges as a bidirectional connection (from the leader to the member and vice versa). This theory claims that the leader does not distribute the time and resources at his disposal equally to all members of his team. Some of his team members (in-group) will deal with the most critical projects and, consequently, will have access to more and more information and interest from the leader, and, correspondingly, the leader forms a high-quality relationship (strong trust-, emotional-, and respect-based relationships) with these members in relation to the rest (out-group) [45,46,47]. By developing mature relationships, leaders and members gradually increase the amount of interaction and reciprocal influences, thus increasing the total amount of resources (i.e., faster execution of a project and, therefore, more time available for new projects) available to the working group. High-quality binary relationships are also characterized by high levels of trust, compatibility, respect, and expectations of cooperation, while the leader, in this case, assigns interesting projects and additional responsibilities to the members he trusts and offers rewards [48].

LMX theory focuses on the leader–member interactions from which qualitative dyadic relationships may emerge where they exchange information of mutual benefit. Those relationships of trust and encouragement have communication channels characterized by openness; they allow mutual influence, generally like working together, and alliances are built [49]. In comparison with in-group relationships, in a typical leader–member (out-group) relationship, the leader does what he typically has to do by sharing the minimum information required to complete the project and so does the member; thus, work results stem from the respective work which is restricted to an order-framework. Meta-analytic studies of the results of quality LMX unveiled that LMX is positively associated with the performance of subordinates, faster hierarchical promotion of employees, employee satisfaction with the leader, organizational commitment, and clarity of job descriptions (work, responsibilities, attitude towards the consumer, etc.) [31,50,51]. All in all, research shows that crises contexts have an overwhelming impact on healthcare professionals while, in particular, physicians who have high-quality relationships experience less burnout and lower overall stress [52,53], and nurses who have high-quality LMX relationships experience high levels of communication, job involvement, and develop positive attitudes towards the organization [48,54].

#### 2.2.2. Organizational Silence

There is an increasing demand in organizations with respect to encouraging employees to speak up, to be freed from the shackles of ‘isolation’ in speaking [55], to engage in a promotive voice meant as a response to challenges of the environment, and to share these efforts through open channels of communication within the organization [56,57,58]. Furthermore, positive and open relationships between leaders and members are of high importance for the future of nursing leadership whereas challenging, lean structural health environments make consistent contact among individuals at all levels difficult [59]. Organizational silence is a complex and multidimensional phenomenon that exists in organizations and concerns behaviors such as withholding ideas, information, and opinions in the workplace; being reluctant or afraid to speak up about certain issues or problems [55,58,60,61]; or even being proactively silent in regard to cooperation [62].

Healthcare organizations are characterized as hierarchical, multifaceted, and complex systems [55,63] where leadership emerges as an individual skill of those who are in leading positions and hold a heroic profile [64]. Research in the past has revealed that the majority of first-line managers did not see their organizations encouraging employees to express ideas, opinions, and information openly [65], and in regard to the bureaucratic culture in healthcare settings, this lack of openness and freedom in speaking up and the suppression of voices is present throughout the system [55]. In contrast to silence, active listening (verbal and nonverbal) enables a deeper connection between the speaker and listener regardless of the type of relationship, while a good listener (e.g., members) enables an environment of ripe cognitive processing of each situation and its coping resources (e.g., coping in crises) [66]. Moreover, silence may also be seen as an informal culture that entails lying between rumors or conversations, so leaders must develop strong network connections to understand employees’ beliefs in-depth and allow words to ‘come to light’ (speaking out) [19] (pp. 65–70) [67]. Silence about important issues may gradually lead to the constraint of the ability of the organization to detect errors and engage in learning through a process of openness in speaking up [68], a climate that one can imagine how harmful it could be in healthcare organizations (e.g., threats to patient safety and care quality) [55,69,70]. The LMX–organizational silence relationship has already been supported by existing studies, yet there is not sufficient research to support this in the healthcare sector [71,72]. All in all, since LMX is a theory that requires interpersonal trust in organizations, it is described as amenable to the vulnerability of specific others and has a pivotal impact on followers’ reactions [72,73]. Because organizational silence is partly related to fear and the suppression of truths, views, and feedback [74], it is assumed that organizational silence and the holding back of opinions will act as an obstacle between leader–member relationships and is perceived as relational dysfunctionality [55]. Moreover, other results show that the reporting of medical errors reduces in the presence of silence whereas there was a willingness to report medical errors when trust-based LMX existed [75]. Taking into account the aforementioned, the following summary proposition is suggested:

**Proposition** **1.**
*The greater the phenomenon of the hesitation of individuals (followers) to speak up in an organization, the lesser the quality of the relationships between the leader–member will tend to be.*


#### 2.2.3. Time

Graen and Uhl-Bien [29] highlighted three different stages in the leader–follower relationship by dividing them into three developing stages: the ‘stranger’ phase, the ‘acquaintance’ stage, and the ‘mature partnership’. The act of dividing relationships based on the maturation processes of each stage demonstrates a timescale process; thus, the three stages have a common denominator and that is ‘time’. As time passes by, leader–member interactions differentiate in size (amount of information and resources exchanged), kind (professional or personal), and reciprocation (contractual, behavioral, and emotional). Schyns et al.’s [76] research results unveiled that relational tenure has no significant correlation with LMX when taking into account the quantitative components of the relationship, which is the total duration of each tenure, i.e., the tenure of the head of the department in relation to the tenure of first-line employees).

Secondly, the paradigmatic assumption of Tesluk and Jakobs [77] suggests that the term ‘work experience’ has a ‘density’ side, meaning that the number of challenging situations an individual is assigned in a specific timeframe is an important parameter for the exegesis of this term. Even more, the ‘frequency’ of communication between team members of a project has been identified as a key for coordination [28]. In the clinical environment, relationship-based communication that involves both-way dialogue (leader and member) and expression of opinions in comparison to task-based communication yields better outcomes [78]. Thereby, it is assumed that if one views leader–member relationships from a qualitative way, then ‘time’ should be measured in terms of the number (‘density’) of quality leader–follower work-meetings during the years the specific relationship lasts. Therefore, the following propositions are suggested:

**Proposition** **2.**
*The negative impact of organizational silence on the quality of LMX relationships is influenced by the duration of the dyadic relationship (total time duration).*


**Proposition** **3.**
*The negative impact of organizational silence on the quality of LMX relationships is influenced by the ‘density’ of work meetings (time per week) between the leader and member.*


#### 2.2.4. Relational Identity

Relational identity is founded upon the scope in which individuals define themselves in terms of their dyadic interactions [79]; that is, relational selves are fundamentally interpersonal and formed, maintained, and displayed through interaction with the specific individual (e.g., family member, coworker, friend, etc.) or the individual others (social groups) [80,81,82,83]. The relational level of self-identity brings about a new perspective of how executives influence the thoughts and actions of subordinates [84]. In leader–member relationships where leaders report strong relational identity, members show high-quality LMX [85], and that is explained through the fact that, at this level, self-worth is based on the value that the specific others place on the relationship. Furthermore, mature leadership relationships are initiated through an investment process of reciprocal interactions between leaders and high-potential members concerning job duties and role responsibilities [86]. Moreover, identities are intertwined with the partnerships, resulting in self-worth being contingent on the welfare of the specific others [79,87]. This co-created relational identity promotes positive attitudes and behaviors [88] because selves reflect on the specific others, and that involves affect and motivation [81]. Therefore, it is argued that these positive attitudes will not be able to ‘spark’ into a system when the dysfunctional climate of silence systematically occurs in organizations.

**Proposition** **4.**
*The more the interpersonal level of interaction between leader and follower enhances the interpersonal bonds and role relationships, the higher quality the relationships between the leader–member will tend to be.*


**Proposition** **5.**
*The negative impact of organizational silence on the quality of LMX relationships will decrease as the relational identity of a relationship gets stronger.*


#### 2.2.5. Trust

Scandura and Pellegini’s [89] research has revealed that trust dynamics are present in high-LMX relationships and that even calculus-based trust, which is based on the value derived from weighing the outcomes based on the benefits of maintaining the relationship and the costs of ending it, exists in high LMX. The level where trust is present, also identified by Graen and Uhl-Bien [29], exists in relationships characterized by high reciprocity [46], and the kind of trust that exists at a low level of LMX is calculus-based trust, which dissipates when the quality of LMX increases [90]. Moreover, trust is considered a significant dynamic not only among LMX relationships but also in research evidence showing that work-related performance, that is, work outcomes of healthcare professionals, can enhance through leadership in the presence of trust [91]. Because employee silence refers to a communicative choice to either suppress or express ideas, perspectives, and opinions on different kinds of topics and thus constitutes a conscious decision of employees [92], it is argued that this conscious withholding of information is also related to a low level of trust (e.g., avoiding speaking up about a medical malpractice event out of fear of retribution) in leader–member relationships [69,93].

Silence is characterized as the conscious decision to hold back seemingly noteworthy information [92], and interpersonal trust is a cognition-based phenomenon (i.e., a choice) and involves a degree of cognitive familiarity of the object of trust that lies between the total knowledge and the total ignorance [94,95]. So, both silence and interpersonal trust are conscious choices. Therefore, it is argued that when the employee consciously chooses ‘silence’, one will not expect to see high levels of ‘trust’ in this relationship. Therefore, following the aforementioned conceptual framework, the following propositions are suggested:

**Proposition** **6.**
*The more employees consciously choose to suppress ideas and withhold information, the lower the level of interpersonal trust within the relationship.*


**Proposition** **7.**
*Organizational silence and its negative affect on LMX, such as reduced trust between the leader and the follower, decrease as the relational identity of the specific partnership is strengthened.*


#### 2.2.6. Job Satisfaction

Job satisfaction is related to the emotional state that stems from the way one experiences work-related situations (i.e., experiences from subordinate communication, corporate information, etc.) [96,97]. To begin with, studies in the past have shown that LMX and job satisfaction are interrelated and have bidirectional positive and significant influence and that job satisfaction predicts the quality of an LMX relationship over time [31,98]. Other research findings have shown that job satisfaction is an important factor for the development of organizational commitment among healthcare workers [99]. Even more, research in the past has shown that high-quality LMX can predict organizational outcomes, such as greater satisfaction with the supervisor, a stronger organizational commitment, and better performance by objectives [100]. So, examining the specific environment in which job satisfaction and LMX evolves is of significant importance and so are the valuable results (e.g., commitment and high communication) in healthcare settings. Furthermore, studies mention that positive job attitudes are correlated with employees contributing drive rather than withholding behaviors [96]. Moreover, satisfaction can be viewed through the lens of an enjoyable and fulfilling communicative experience, and in organizational settings, satisfaction is defined by interconnectivity and interaction with the others and the context within which one operates [97].

The literature implicates active links between LMX and job satisfaction. For instance, in high-quality LMX (in-group members) the leader and member have an open ‘channel’ of communication and build alliances, the leader shares important information, and members do tend to have faster work promotability, cues that are positively related to job satisfaction [29,44,90,98,101]. Consequently, it is argued that a rich flow of communication and openness in the leader–member relationship is enhanced by both quality LMX and job satisfaction. In that sense, it is highly expected that organizational silence, which is considered a reluctance to speak up about significant issues of concern [58], will negatively affect the dimensions of LMX and job satisfaction in an organizational setting. Therefore, following the aforementioned conceptual framework, the following propositions are suggested:

**Proposition** **8.**
*The more employees suppress ideas and withhold information, the lower job satisfaction will they experience.*


**Proposition** **9.**
*LMX is positively related to job satisfaction, and this relationship is negatively influenced by organizational silence.*


#### 2.2.7. Organizational Citizenship Behavior (OCB)

OCB is discretionary in nature behavior that employees display when exceeding their formal job role, constituting extra-role behavior, and is an added value to every organization [102]. LMX could not be examined separately from organizational outcomes since the level of quality of LMX in relational terms determines critical organizational outcomes [48,103]. Research in the past has shown that the high quality of leader–member relationships is associated with the development of the followers’ role. Meaning, a rich flow of communication, influence, and support are characteristics that constitute exceeding job descriptions and work contracts [48,51]. Firstly, other studies in the past indicate that when individuals are experiencing a high quality of exchanged experiences and reciprocation with their leaders, then they feel more attached to the organization as a whole [48,104]. As a consequence, they increase in-role performance and responsibility to promote a beneficial change in their organization [57]. Secondly, other studies reveal that when employees are emotionally disconnected from others in organizations, then they are more likely to reduce extra-role behaviors [104]. Emotional disconnection negatively influences employee voice, resulting in the hiding of thoughts, feelings, creativity, and beliefs [104,105].

Moreover, a rich exchange of information enhances participation in decision making, promotes a sense of purpose for employees, creates channels of feedback, and allows for a ‘giving–taking’ relationship to happen in order to support organizational goals and results [59]. Additionally, studies show that where high LMX and trust are present, medical incident reporting is illustrated as extra-role behavior [106]. Moreover, shifting decision making from managers to subordinates stimulates coworkers’ emotional support and guidance to peers in achieving a higher level of performance, which means that when leaders unleash participation dynamics (i.e., give ‘voice’ to employees’ opinions) and allow helping behavior to happen, then citizenship-like behaviors arise [107,108]. Driven by the fact that, by virtue of employee voice, both LMX exchanges and OCB behaviors are enhanced, it is argued that the absence of employee voice (i.e., organizational silence) will negatively intercede this relationship.

**Proposition** **10.**
*Qualitative relationships between the leader–member and their interrelatedness with OCBs will be mediated by the level of speaking ‘isolation’.*


## 3. Discussion

A number of implications may be derived from this conceptual framework of relational leadership theory in healthcare organizations when nowadays, system unpredictability is more than obvious. In this study, seven propositions were displayed to be explored regarding the context of relational interactions (leader–member dyads) where relational processes occur and by which relational leadership is ‘born’ and enabled, thereby ‘stretching’ the results (outcomes) [42]. Additionally, three propositions were suggested as occurring in the space of relational outcomes by viewing leadership as an outcome (e.g., job satisfaction and OCB) (Figure 1).

This theoretical analysis attempts to highlight the importance of paying keen attention to the quality of leader–member relationships in a broader relational mindset of how things are correlated (positively or negatively), synthesizing the specific relational puzzle that will allow smooth adaptation (administratively and operationally) and further production of positive outcomes within a constantly changing environment [109]. Operational processes in healthcare should rely on relationship-based communication because when working in an operating room and one of the team members’ notices something is not proper, he or she should feel that there are no negative repercussions for speaking up [78].

Furthermore, this study’s analysis regarding ‘time’ on a daily basis in an organization can draw affiliation clues from the paradigm of the relationship of mother–infant where time (hours per day) of the mother’s engagement with the infant is an indicator of positive mothering and the creation of a quality home environment for the child [110], implying that ‘time’ is a crucial factor both in the mother–child relationship and in the leader–member relationship. In addition, in a positive mother–infant relationship, the mother develops maternal sensitivity and responsiveness to the needs of the infant, and the infant shows commitment and positive response in their relationship exchanges [110,111]; hence, it is a challenge to further explore shared ‘time’ in LMX exchanges.

Attention should also be paid to the fact that frequent communication keeps healthcare employees updated on patients’ progress, raises familiarity with med-issues, and builds relationships [28]. As a result, during the day or week (short term), supervisor–subordinate contact is likewise important. The need to enhance the quality of communication in the leader–member relationship implies the need to further explore the influence of the agents involved. That is, supervisors may often think they are seeing the big picture of the organizational environment while the opposite may be happening, such as how supervisors may not have awareness of the ‘tendency’ towards silence. In addition, managers may tend to assume that ‘no news is good news’, meaning that they think followers feel free to express their opinions in any case, which is not what actually happens in many cases. Therefore, future research on this field will help improve the channels of communication in LMX relationships which, in turn, will help peers of the successors to build stronger ties with their leaders (e.g., empower trust, loyalty, etc.), promote ‘workplace’ friendship, which foreshadows superior performance [112] (pp. 242–244), and learn valuable lessons (e.g., how to challenge the status quo in times of market recession) through this solid relationship [58,113].

By virtue of the fact that research has indicated that a high percentage of healthcare workers do not speak up regarding important problems, such as ethics, responsibilities, administrative problems, or other employees’ behaviors, this study features the exploration of this phenomenon throughout the relational context of healthcare organizations [69,114]. Additionally, since relationship-oriented employees regulate their behaviors by not having fears around the achievement of goals and are not afraid to challenge partnerships [85], field leadership should be targeted in that direction. Even more, seeing organizations and relationships through the lens of relational identity is a field not yet broadly explored by researchers [84]. Considering that studies have displayed that interpersonal relationships are salient with regard to organizational commitment and job performance [107], it is more than obvious that examining relational leadership will shed light on practical implications for supervisors and organizations.

Following the aforementioned, three practical relationships between leader and member are suggested in this study as a steppingstone towards enhancing leadership development:In light of the fact that interpersonal relationships can be increased [44] and that research findings show that employees who are strongly trusted by their leaders experience a better quality of LMX with their leader [115], leaders are urged to focus on demonstrating high levels of trust towards their employees.Leaders are encouraged to pay attention to the quality of their relationships along with enhancing ‘voice’ in their teams since research results show that LMX in healthcare employees is positively related to incident reporting, a fact that is so crucial in healthcare settings in terms of patient and staff safety [106].Leaders are urged to create channels of open communication with their followers given the fact that studies show that the frequency of communication keeps professionals updated on patient progress and enhances LMX [27,54,78], and secondly, having organizational silence is related to decreased OCB, that is, with decreased outcomes [116].

## 4. Conclusions

The present paper extends the understanding of leadership development by viewing leadership from the standpoint of interactions (leader–member), rather than just from the standpoint of the leader or the follower as separated individuals [24,117,118], together with trying to morph a broader framework on how relational leadership is constructed in healthcare organizations by describing the forming (context of interaction), enabling, and emerging (relationships as outcomes) of leadership [7]. Leadership theory today is in a post-heroic phase and exploits its dimensions beyond focusing only on leaders [119] towards any level of connectivity between actors in the workplace in order to manage complexities in healthcare [120]. Nonetheless, healthcare organizations are still strongly hierarchically structured [19,121]; thus, this research unfolded the challenges at the leader–member level of communication [122], underpinning the importance of focusing on both sides of an interaction and on the effect of relational dynamics emerging through those interactions in order to optimize outcomes.

The current analysis suggests at least four directions for future research. First, given that healthcare organizations need to adapt to constantly changing, complex environments by having readiness [123] and situational awareness in adverse working conditions [124,125], the importance of focusing on human-centered leadership methods is emerging [67]. Second, given that healthcare organizations are networks of rich human interconnectivity and individuals usually have multiple relational identities (e.g., with colleagues, subordinates, managers, patients, etc.) [1], exploring leader–member relationship is the beginning of the unfolding the social process that influences the effectiveness of healthcare organizations. Third, the implications of this socially emerged climate of ‘silence’ within healthcare organizations is a crucial domain to be explored, drawing from the fact that ‘silence’ was intensified within the context of economic crisis [61] in other type of organizations. In parallel, Harris and Gresch [126] (pp. 195–196) reported that ‘voice’ in organizations is, rather, an attempt to change a situation and not refrain from it; hence, it is important to dwell in a positive environment that allows employees to speak up [109]. Fourth, exploring the potential of RLT and how leadership is enabled is a steppingstone towards enhancing a problem-solving mindset rather than a blaming mindset (e.g., mindset concerning medical errors) where the latter is a regular phenomenon in healthcare settings and, unfortunately, leads to sweeping information under the carpet [28,55,127].

Moreover, little research is done recently regarding relational dynamics (e.g., leaders’ silence, ‘wise listening’ by members, etc.) at the individual level of interactions, and there is still room for understanding how those dynamics coexist at the collective level (e.g., intergroup silence) in healthcare organizations [72,128,129]. Due to the permeating presence of the aforementioned dynamics in healthcare environments [55,120], it is highly recommended future researchers access those dynamics [130] to add to the value of leadership development.

Finally, drawing from the fact that the leadership field is still not sufficient for explicitly addressing the issue regarding levels of analysis (individuals, dyads, larger collectivities, etc.) [7,43], this study aimed at contributing to the next evolution stage, which involves the processes and phenomena beyond leader–member interactions and moves towards the relational dynamics which exist and are produced throughout the workplace.

## Figures and Tables

**Figure 1 ijerph-19-15971-f001:**
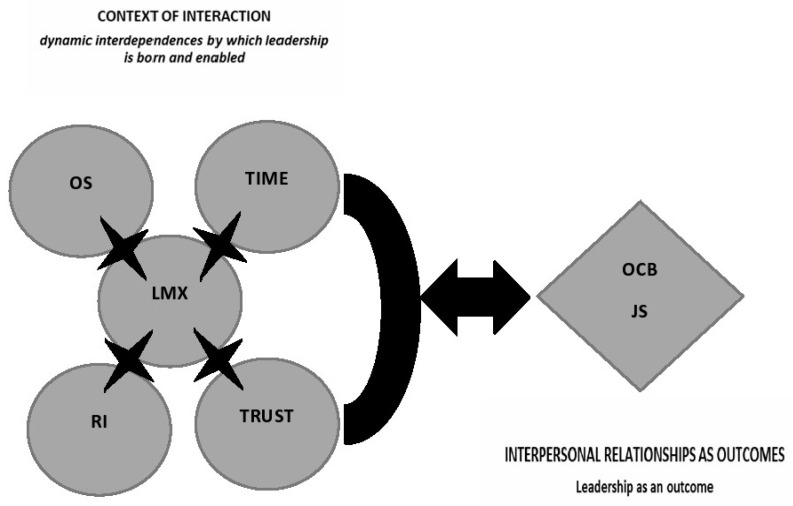
Interrelated dynamics by which relational leadership is developed and enabled, stretching the outcomes. LMX: Leader–Member Exchange; RI: Relational Identity; OS: Organizational Silence; JS: Job Satisfaction; OCB: Organizational Citizenship Behavior.

## Data Availability

Not applicable.

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
