# Peer review of "Leadership Development: Exploring Relational Leadership Implications in Healthcare Organizations"

_ijerph, 2022, doi:10.3390/ijerph192315971_

Round 1
Reviewer 1 Report
1) Please revise the writing of:
- “e.t.c.” (Line 142). The correct form is “etc.”
- “Behaviour” (Line 332). You must use the American English for this word, i.e. “Behavior”
- “On the other hand” (Line 192). This is an informal expression. The proper word must be “Secondly”
- “On the one hand” (Line 294). Please use “Firstly”
- “On the other 298 hand” (Lines 298-299). ). Please use “Secondly”
2) The authors should:
- add the results of a research in which the correlations mentioned in the 10 prepositions were tested.
- or reorganize the paper with fewer prepositions in which all the correlations were tested through a research.
Reviewer 2 Report
This study explores both the dyad relationships and the context in which those occur. With recent attention directed to the implementation of human-centered practices, the creation of effective networks, and to bringing desired results, the RLT is called on to advance this agenda within healthcare organizations. However, english-only research articles that examined leadership theories over the past thirty years were selected from computerized databases and manual searches. It is noticed that the way relationships are formed between leaders and members and in a specific social context is a social process that in turn shapes the effectiveness of management of those organizations. Leadership is not rank - it is the relationships with the relational dynamics that play in the same context creating evolutionary organizational processes.
The novelty of the article must be highlighted.
Why only English articles were considered?
Section 2 must be shortened.
Future work should be mentioned in the conclusion section.
Reviewer 3 Report
Dear Authors,
An interesting topic has been covered, based on the amount of reviewed literature, it can be seen that you covered the topic well, extensively and from various angles.
What led you to this topic? covid-19?
Did you manage to conclude what is common to the individual and the society when, for example, there was Sars, measles, plague or covid-19? And with some conclusion to imagine what would happen if a new strain of virus happened in the future?
Why didn't you state in your paper some concrete, general and practical relationship between leaders and members in healthcare?
In paragraph 143, you mentioned nurses, why didn't you mention somewhere the relationship between the leader of the doctor and the member of the doctor, and then the relationship of the whole team, for example, the department for cardiovascular diseases?
In paragraph 177, proposition 1, have you considered what happens when the leader speaks less or more? How does it affect other members and in what capacity?
Do members who "listen wisely" have better success with leaders than those who are too open and talk too much?
You should have inserted healthcare in the time and trust section
You should have introduced LMX more into the concept of healthcare because that is the title of your paper
In my opinion, the work should have been colored a little with another sketch, graphic, picture. This somehow seems dry and gloomy for other people to read
The conclusion must be more specific.
